# The Retrospective Study of Magnetic Resonance Imaging Signal Intensity Ratio in the Quantitative Diagnosis of Temporomandibular Condylar Resorption in Young Female Patients

**DOI:** 10.3390/jpm13030378

**Published:** 2023-02-21

**Authors:** Shaonan Wan, Qi Sun, Qianyang Xie, Minjun Dong, Zhiyang Liu, Chi Yang

**Affiliations:** 1Department of Oral Surgery, Shanghai Ninth People’s Hospital, Shanghai Jiao Tong University School of Medicine, Shanghai 200011, China; 2College of Stomatology, Shanghai Jiao Tong University, Shanghai 200011, China; 3National Center for Stomatology, Shanghai 200011, China; 4National Clinical Research Center for Oral Diseases, Shanghai 200011, China; 5Shanghai Key Laboratory of Stomatology, Shanghai 200011, China; 6Department of Radiology, Shanghai Ninth People’s Hospital, Shanghai Jiao Tong University School of Medicine, Shanghai 200011, China

**Keywords:** temporomandibular joint, anterior disc displacement, magnetic resonance imaging, condylar resorption, signal intensity ratio

## Abstract

According to the literature, there is no reliable and quantitative method available for the diagnosis and prognosis of early or potential temporomandibular joint (TMJ) condylar resorption (CR) thus far. The purpose of this study was to raise a new noninvasive method to quantitatively evaluate condylar quality using the signal intensity ratio (SIR) on magnetic resonance imaging (MRI) in order to assist in the diagnosis of TMJ CR. A retrospective exploratory study was performed to compare the condyle-to-cerebral cortex signal intensity ratios (SIR) on MRI among young female patients. We included 60 patients, and they were divided into three groups: the bilateral normal TMJ group (group 1), the bilateral TMJ anterior disc displacement (ADD) but without CR group (group 2), and the bilateral TMJ anterior disc displacement (ADD) with CR group (group 3). The SIR difference between the three groups was analyzed by the Kruskal–Wallis test (K-W test). The sensitivity, specificity, accuracy, and area under curve (AUC) were calculated by the receiver operating characteristic (ROC) curves. There was high consistency between the surgeon and the radiologist in the evaluation of the magnetic signal intensity with intraclass correlation coefficients of 0.939–0.999. The average SIR was 1.07 in the bilateral normal TMJ group (group 1), 1.03 in the ADD without CR group (group 2), and 0.78 in the ADD with CR group (group 3). It could be found by the K-W test that group 3 was significantly different from group 1 and group 2 (*p* < 0.05), while there was no significant difference between group 1 and group 2. The optimal critical SIR value was 0.96 for the diagnosis of CR according to the ROC curves and Youden index (*p* < 0.001, AUC = 0.9). The condyle-to-cerebral cortex SIR can be used as a noninvasive diagnostic tool for the quantitative evaluation of condylar quality and diagnosis and prognosis of CR. SIR ≥ 0.96 indicates a healthy condyle, while SIR < 0.96 is considered the optimal critical value for the diagnosis of CR. These findings are important for personalized and accurate treatment and prognosis prediction.

## 1. Introduction

Condylar resorption (CR) occurs in conditions that cause osteolysis and volume loss of the condyle [1]. CR can also lead to mandibular deformity, malocclusion, temporomandibular joint (TMJ) and mandibular dysfunction, pain, and headache [2]. There are many different TMJ pathologic abnormalities that can cause CR. According to our previous study, TMJ anterior disc displacement (ADD) might have a strong relationship with CR [1,3,4]. The prevalence of anterior disc displacement (ADD) of TMJ is as high as 30–40%, especially in adolescents and young adult females [5], occurring more frequently than TMJ posterior, medial, or lateral dislocation [5,6]. According to clinical observation, the occurrence of simple TMJ lateral displacement is rare, usually being asymptomatic and not requiring any treatment. Therefore, most of the academic vision has been focused on “ADD patients”.

In clinical examination, disc displacement is often diagnosed by examining the patient’s mouth opening degree and whether there is clicking or limitation when opening mouth [7,8,9]. However, the diagnosis of clinical examination tends to lack imaging evidence. In imaging examinations, a study on the research diagnostic criteria for temporomandibular disorders (RDC/TMD) demonstrated that, using these criteria, the reliabilities of radiologists in the assessment of osseous diagnosis with computed tomography (CT) and disc diagnosis with magnetic resonance imaging (MRI) were good [4]. Thus, for diagnosis of TMJ ADD, MRI is an effective tool for diagnosis of TMJ ADD due to its capacity to clearly display the morphology of the TMJ disc, condyle, and glenoid fossa and their relative positions, and it is also safer than CT due to the lack of ionizing radiation [10]. Thus, MRI has been widely used for the diagnosis of TMJ diseases, including anterior disc displacement, idiopathic condyle resorption, and synovial chondromatosis. However, the diagnosis and prognosis of CR is often difficult to make due to the pronounced individual differences and the lack of universally accepted diagnostic tools. 

Apart from structure changes of the condyle, the quality of it also affects the prognosis of the disease. The blood supply of the condyle plays an important role in condylar growth, nutrition supply, and repair potential. Long-term displacement of the articular disc can lead to decreased blood supply to the condyle and consequently the occurrence and development of CR [11]. Therefore, the information of blood flow in the condyle can be useful in the diagnosis and prognosis of CR. In the proton density weighted imaging (PDWI), the magnetic signal intensity (MSI) can roughly reflect the blood flow signal. However, due to differences in individual attributes and filming conditions when MRI is taken, the absolute MSI of the condyle has no clear reference value. Some scholars found that the MSI of the cerebral cortex was a good reference for quantitative MRI analysis because of its high repeatability and accuracy, and it was maintained within a certain range for a given patient [12]. Therefore, by calculating the ratio, the MSI of the condyle can be standardized on the basis of the MSI of the cerebral cortex.

For ADD patients, it is important to evaluate condyle quality and the risk of CR by using non-invasive and quantitative methods in order to formulate an accurate and personalized treatment plan, analyze prognosis, and avoid potential bone resorption risk. 

The purpose of this study was to raise a new noninvasive method to quantitatively evaluate condylar quality by using the signal intensity ratio (SIR) on MRI to assist in the diagnosis of TMJ CR. We hypothesized that assessment of the condyle-to-cerebral cortex SIR on MRI as a noninvasive and quantitative method is very important in early diagnosis and prognosis of CR.

## 2. Methods

Ethical Statement: The authors are accountable for all aspects of the work in ensuring that questions related to the accuracy or integrity of any part of the work are appropriately investigated and resolved. The study was conducted in accordance with the Declaration of Helsinki (as revised in 2013). The study was approved by the Independent Ethics Committee of the Ninth People’s Hospital (SH9H-2019-T259-3). 

### 2.1. Patients

Sixty female patients aged from 14 to 18 who underwent MRI of TMJ in the Shanghai Ninth People’s Hospital from July 2019 to July 2020 were collected. Of these 60 patients, 20 patients had a bilateral normal disc–condyle relationship, and condylar morphology was included in the normal group (group 1), with the other 40 patients being bilateral ADD patients who had received clinical follow-up treatment and underwent MRI examination one year apart to observe the disc–condyle relationship and condylar morphology. According to the clinical symptoms, such as mandibular retrognathia and anterior opening bite, as well as the comparison between MRI results before and after, 20 bilateral ADD patients had no obvious clinical symptoms, and condylar morphological changes were included into ADD without CR group (group 2), and 20 bilateral ADD patients who had obvious clinical symptoms and condylar morphological resorption or loss of height after one year of follow-up were included into the ADD with CR group (group 3). Inclusion and exclusion criteria were as follows: Those included in group 1 ① were 14–18 years of age; ② had no joint symptoms and received MRI as required by orthodontic treatment; ③ had a normal bilateral disc–condyle relationship and condylar morphology on MRI. Those included in the group 2 ① were 14–18 years of age; ② had some clinical symptoms for bilateral joints, such as clicking, pain, and limited mouth opening; ③ had bilateral ADD but normal condylar morphology after one year follow-up on MRI. Those included in group 3 ① were 14–18 years of age; ② had bilateral ADD on MRI and had some obvious clinical symptoms such as mandibular retrognathia, limited mouth opening, and open bite of anterior teeth; ③ had obvious condylar morphological resorption and loss of height on MRI after one year follow-up. 

Cases were excluded if they ① had abnormal MRI signal in the brain or condyle; ② had systemic diseases of bone metabolism; ③ had metal objects in the maxillofacial region that could lead to large-area artifacts in MRI and CT images; or ④ had trauma, tumor, or serious infection involving the mandible and TMJ, such as jaw osteomyelitis.

### 2.2. MRI

All patients underwent magnetic resonance scanning of bilateral TMJ with a 3.0T magnetic resonance scanner (Ingenia; Philips Healthcare Systems, the Netherlands) following the standardized protocol [13]. The three sequences were the closed oblique coronal T2 weighted imaging (T2WI), closed oblique sagittal proton density weighted imaging (PDWI), and open oblique sagittal T2WI. 

### 2.3. Measurements and Analysis

The MRI images were processed by ProPlan CMF software (version 1.4, Materialise, Leuven, Belgium). Three MRI images were obtained at the middle third of the condyle at each side, where the cross-section was largest. The MRI of the condyle was performed at the condylar head and the inner condylar cortex, and the largest inscribed circle in each slice was used for sampling. Measurement was repeated three times for each side. For the cerebral cortex, MRI was performed at the cerebral cortex at the inferior temporal gyrus located at the same position of the condyle. Three measurements were taken for the inscribed circle of the same area for each side, and the average was reported. The MSI of the condyle (MSI1) and cerebral cortex (MSI2) were obtained, and then the SIR (MSI1/MSI2) between them was calculated. Samples were presented as follows (Figure 1A–C). A total of 120 condyles were measured independently by a surgeon and a radiologist.

### 2.4. Statistical Analyses 

All statistical analyses were performed using SPSS v.21.0 (IBM Corp, Armonk, NY, USA), PASS (v.13.0.6, NCSS, LLC, Kaysville, UT, USA), and STATA (v.14.2, 2015, StataCorp, College Station, TX, USA). The intra-group correlation coefficient (ICC) was calculated to evaluate the consistency between the surgeon and the radiologist. All data were tested for normality by the Shapiro–Wilk test. If the data were normally distributed, analysis of variance (ANOVA) was used, and the Kruskal–Wallis test (K-W test) was performed to compare the SIR difference among three groups if the data were not normally distributed. *p* < 0.05 was considered statistically significant. Sensitivity, specificity, accuracy, and area under the receiver operating characteristic (ROC) curves were calculated on the basis of histological reference standards. The Youden criteria were used to establish the cut-off point with AUC > 80%. The condyle at each side was taken as the analysis unit.

## 3. Results

### 3.1. Consistency of Measurements

The MSI data of the cerebral cortex and condyle were reviewed independently by a surgeon and a radiologist, and the consistency between them was evaluated. It should be noted that the ICC values ranged from 0 to 1, where 0 indicates completely unreliable, and 1 indicates completely credible. The reliability coefficients lower than 0.4 indicate poor reliability, while those greater than 0.75 indicate good reliability. It was found that the consistency was very good (Table 1). 

### 3.2. Comparison of SIR Values among Three Groups

The date of group 1 was not normally distributed. In group 1 and group 2, the MSI of the condyle was equivalent to or slightly higher than that of the cerebral cortex. Therefore, SIR tended to be equal to or greater than 1. However, the MSI of the condyle was significantly lower than that of the cerebral cortex in group 3. Therefore, SIR tended to be less than 1 (Table 2). Comparison of SIR values among three groups by the K-W test were as follows (Table 3): From the results, it could be concluded that group 3 was significantly different from group 1 and group 2 (*p* < 0.05), while there was no significant difference between group 1 and group 2.

### 3.3. Analysis of the ROC Curve and the Optimal Critical Value of SIR

The AUC of the ROC curve was 0.945, the 95% CI was 0.888–0.979, and the standard error was 0.0204 (*p* < 0.001) (Figure 2). The optimal critical value of SIR was 0.96 according to the Youden index (SEN + SPE-1). The sensitivity was 90%, and the specificity was 88.75%.

## 4. Discussion

### 4.1. Quantitative Evaluation of Condylar Quality Based on TMJ MRI

At present, MRI is considered the gold standard for the diagnosis of ADD and an important means of evaluating CR. Many scholars believed that CT and X-ray are also good means to examine the condyle [14]. X-ray examination was not preferred to be the first choice because of the problem of artifacts and the inability to observe condylar shape in three dimensions [15]. Although CT was able to better display the cortical bone morphology of the condyle, its imaging of soft tissue, especially the articular disc, was not as clear as MRI, and it might also cause radiation injury [16]. Therefore, we chose MRI for routine examination of the condyle of all patients. Previously, professor Yang Chi proposed a new staging method (Yang’s staging) to guide the treatment plan and prognosis evaluation according to the position of the articular disc and the integrity and morphology of the condylar bone cortex on MRI [17]. However, current diagnosis of CR and risk prediction mainly relies on the doctors’ experience, and the lack of quantitative diagnosis standards makes it difficult to formulate treatment plan and prognosis prediction.

According to the principle of MRI, active protons are excited by the magnetic field to produce resonance and signal echo. Therefore, high-density signals will be produced by tissues that are rich in fat and water [18]. The condyle has abundant blood supply and contains a certain amount of fat, and thus the signal formed by water molecules in the condyle medullary cavity reflects the blood perfusion and fat content. The MSI is reflected in the form of gray value on MRI. The MSI of the cerebral cortex has been used as the reference for quantitative MRI research because of its high repeatability and accuracy, and it is maintained within a certain range for a given patient. Therefore, we propose that under standardized examination conditions, the MSI of the condyle is likely to be proportional to that of the cerebral cortex in subjects with normal TMJ. Thus, changes in the condyle quality in ADD patients may change the condyle-to-cerebral cortex SIR, and the condyle will be at risk of resorption when the SIR is lower than the normal range.

### 4.2. Relationship between SIR and CR

According to the K-W test of this study, it was found that the condyle-to-cerebral cortex SIR was relatively stable (1.07 ± 0.13) in patients with normal TMJ (group 1) under the same the MRI parameters. For patients who have TMJ ADD but without CR (group 2), SIR decreased slightly to 1.03 ± 0.14, which showed no significant difference with that of group 1, indicating that merely ADD would not affect the change of SIR. By contrast, SIR decreased widely to 0.78 ± 0.14 in patients with ADD and CR (group 3), which showed significant differences with the other two groups. From the results above, it could be concluded that SIR tended to decrease with the happening of CR, which could be used as a method to quantitatively evaluate CR. Meanwhile, we combined group 1 and group 2 to form a no-resorption group, and group 3 was the resorption group. ROC analysis was performed for these two groups. According to ROC curve analysis, the AUC of ROC curve was 0.945 (*p* < 0.001). The optimal critical value of SIR was 0.96 according to the Youden index (SEN + SPE-1). At this time, the sensitivity was 90%, and the specificity was 88.75%. That was to say, SIR< 0.96 could be considered the optimal critical value for the diagnosis of CR, while SIR ≥ 0.96 indicated a healthy condyle, providing us with an accurate index for the quantitative evaluation of the condyle.

### 4.3. Possible Reasons for the Decrease in SIR in Patients with CR

We previously found that ADD was sometimes accompanied by CR [19,20,21,22]. In addition, we found that most of the patients with ADD and CR were young female patients [23], especially those in the stage of growth and development, which was consistent with the population and age of idiopathic condylar resorption (ICR). This was the reason we chose 14–18-year-old female patients for our study. However, the cause of CR by disc displacement is still controversial. Some scholars believe that it is related to the change of joint pressure [23,24,25]. The force on the condyle is significantly higher in ADD patients than in subjects with a normal disc–condyle relationship [19]. The literature review suggested that blood supply of the condyle plays a very important role in the growth and development, nutrition supply, and repair potential. The high pressure could inhibit the development of bone and blood vessels [26]. It was also found that the decrease in blood supply would affect the synthesis of aminopolysaccharide in cartilage and cause osteoarthritis, which could impact the growth of condyle bone. Meanwhile, long-term ADD would reduce blood supply in the condyle, slow down blood flow, and affect bone metabolism in ADD patients [27]. Therefore, one possible reason for the decrease in SIR in patients with CR was that the blood supply in their condyle was significantly lower than that in the normal condyle. Moreover, we found that ICR patients tended to be thinner in our experience, and some scholars found that abnormal lipid metabolism could lead to the decline of osteogenesis [28]. Thus, the other possible reason for the decrease in SIR in CR patients might be related to the decrease in lipid signal intensity in the condyle. In conclusion, MSI of the condylar could roughly reflect the blood flow and lipid signal that tended to decrease when CR happened. Thus, a better understanding of the blood flow and lipid metabolism in the condyle is important for the treatment and prognosis of TMJ CR.

### 4.4. Future Prospects

In terms of our future expectations, we hope to raise a “SIR degree” to formulate personalized treatment. When SIR > 0.96, it is suggested that the condyle is under a low risk of CR, and orthodontic treatment can be routinely performed. When 0.78 < SIR < 0.96, it should be careful that condyle is under a high risk of CR. At this stage, prevention of potential CR risk and health education are recommended. If the patient is undergoing orthodontic treatment, attention should be paid to the size and direction of orthodontic force to avoid exerting more pressure on the condyle. Meanwhile, the follow-up of the condyle by MRI is also important for early detection of CR. When SIR < 0.78, it is indicated that CR has already happened or will certainly happen soon. At this stage, measures should be taken to delay the progression of CR, such as appropriate orthodontic treatment to relieve pressure on the condyle [29,30]. Besides this, arthroplasty may be performed if necessary. It has been observed that arthroplasty can effectively slow the progression of CR [22,31]. We hope these suggestions will help clinicians make the correct decision and avoid the risk of CR at different stages. 

On the other side, despite the frequent occurrence of mandibular retraction or deflection, a large number of patients are actually not aware of the state of their TMJs. Even for patients who visit the hospital for the treatment of TMJs diseases, clinicians often tend to spend much energy to track their conditions. Moreover, CR is often caused by missing the best treatment time. It is expected that the diagnosis can be made in a more intelligent way with the introduction of an artificial intelligence (AI) model technology of MRI to finish the measurement of SIR automatically, which will not only greatly reduce labor cost and prompt the risk of CR in time but also can be helpful for patients with no access to precision medical treatment. We believe that SIR can be used for the diagnosis and prognosis of ADD patients with CR. In the future, we will increase the sample size and introduce the AI model technology of MRI in order to assist the quantitative evaluation of CR. 

In addition, in order to make our study more convincing and the number of cases much larger, we are collecting MRI data from a group of patients with unilateral CR. It is hoped that their bilateral SIR values can be self-compared, so as to further verify the significance of SIR values. 

## 5. Conclusions

The condyle-to-cerebral cortex SIR can be used as a noninvasive diagnostic tool for quantitative evaluation of condylar quality and diagnosis and prognosis of CR. According to our study, it was found that the condyle-to-cerebral cortex SIR was 1.07 ± 0.13 in patients with normal TMJ. For patients who have TMJ ADD but without CR, SIR decreased slightly to 1.03 ± 0.14. By contrast, SIR decreased widely to 0.78 ± 0.14 in patients with ADD and CR. Compared with ADD without CR patients and normal patients, SIR in ADD with CR patients tends to be significantly decreased. According to our ROC results, SIR ≥ 0.96 indicates a healthy condyle, while SIR < 0.96 is considered the optimal critical value for the diagnosis of CR. These findings are important for personalized and accurate treatment and prognosis prediction. In terms of our future expectations, we will introduce AI technology in automatic MRI diagnosis, which will help clinicians predict the risk of CR more effectively and intelligently.

## 6. Patents

This section is not mandatory but may be added if there are patents resulting from the work reported in this manuscript.

## Figures and Tables

**Figure 1 jpm-13-00378-f001:**
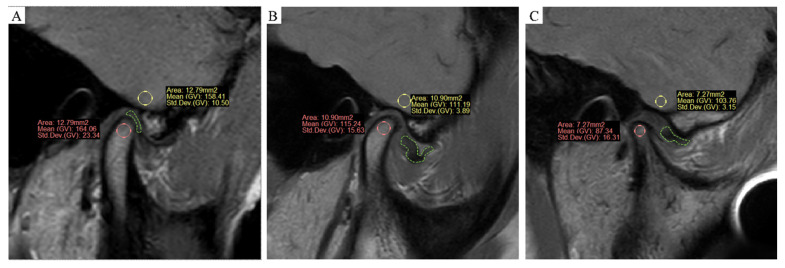
(**A**–**C**) MRI sampling and measurement, oblique sagittal at closed mouth PDWI. (**A**) Normal group (group 1): the articular disc marked in green was in a normal position, and the status of the condyle was healthy. (**B**) ADD without the CR group (group 2): ADD occurred, but the status of the condyle was still healthy. (**C**) ADD with CR group (group 3): not only ADD occurred, but the shape and volume of the condyle was significantly resorbed.

**Figure 2 jpm-13-00378-f002:**
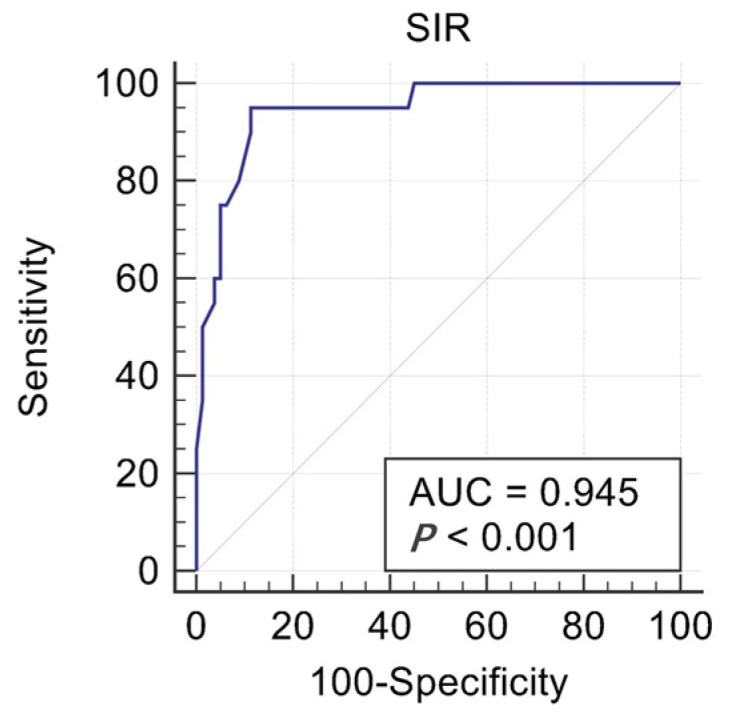
The ROC curve of the SIR value.

**Table 1 jpm-13-00378-t001:** Consistency between the surgeon and the radiologist.

Region of MSI	ICC	95% CI
Right cerebral parenchyma	0.997	0.991~0.999
Right condyle	0.985	0.962~0.994
Left cerebral parenchyma	0.999	0.997~0.999
Left condyle	0.939	0.853~0.975

ICC, intraclass correlation coefficient; CI, confidence interval.

**Table 2 jpm-13-00378-t002:** SIR values in the three groups.

	Min	Max	Avg	Sd	Md
Normal group (Group 1, *n* = 40)	0.95	1.29	1.07	0.13	1.04
ADD without CR group (Group 2, *n* = 40)	0.96	1.20	1.03	0.14	1.01
ADD with CR group (Group 3, *n* = 40)	0.93	0.43	0.78	0.14	0.79

Min, minimum; Max, maximum; Avg, average; Sd, standard deviation; Md, median; ADD, anterior disc displacement; CR, condylar resorption.

**Table 3 jpm-13-00378-t003:** Comparison of SIR values among three groups by Kruskal–Wallis test.

	Group 1	Group 2	Group 3	*X* ^2^	*p*
SIR	1.085 (1.020, 1.128)	1.070 (1.013, 1.070)	0.795 (0.673, 0.920) ^&#^	63.010	<0.001

^&^: There was significant difference between group 3 and group 1, *p* < 0.001. ^#^: There was a significant difference between group 3 and group 2, *p* < 0.001.

## Data Availability

No new data were created or analyzed in this study. Data sharing is not applicable to this article.

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
