# Peer review of "The Retrospective Study of Magnetic Resonance Imaging Signal Intensity Ratio in the Quantitative Diagnosis of Temporomandibular Condylar Resorption in Young Female Patients"

_jpm, 2023, doi:10.3390/jpm13030378_

Round 1

Reviewer 1 Report

Dear Authors the article is really interesting, well conducted 

I suggest to 

-About the Title of the article, I suggest you to modify it and add the type of article.

- The introduction section is very short and is needed to add other references to increase the quality of the manuscript suggested: [https://doi.org/10.3390/app122312409];  [DOI: 10.1155/2022/7091153] ;  [https://doi.org/10.3390/prosthesis4020025] ; [10.1080/08869634.2022.2126079]

-You need to review the grammar of  your paper

-I suggest you add a table with the list of abbreviations used in the text.

-Please expand conclusion section with main results and future perspectives of this study

Kind Regards

Author Response

Response to Reviewer 1 Comments

Dear reviewer:

  Many thanks for the insightful comments and suggestions of the referees. Those comments are all valuable and very helpful for revising and improving our paper, as well as the important guiding significance to our researches. About these comments, we have sepearated our responses point-by-point as follows.

Point 1: About the Title of the article, I suggest you to modify it and add the type of article.

Response 1: This is a retrospective research article. After discussion , we have modified the article’s title. The new title of the article is “A retrospective study of magnetic resonance imaging signal intensity ratio in the quantitative diagnosis of temporomandibular condylar resorption in young female patients”.

Point 2: The introduction section is very short and is needed to add other references to increase the quality of the manuscript suggested.

Response 2:We have enriched the content and references in the introduction section, such as further information on CR, risk factors and characteristics of the patients. Besides that, we rearranged the sequence of sentences hoping to make the article more logical.

Point 3: You need to review the grammar of  your paper

Response 3: We have already reviewed the grammer of our paper sentence by sentence and made some changes in the new version.

Point 4: I suggest you add a table with the list of abbreviations used in the text

Response 4: We have added a table with the list of abbreviations in the “Supplementary Materials” part at the end of our new version.

Point 5: Please expand conclusion section with main results and future perspectives of this study

Response 5: We have added main results and future plan on artifical intelliigence which will be used in our future work to the conclusion section at the end of the paper.

Reviewer 2 Report

Despite the unique topic, there are a few suggestions to improve the clinical impact of the paper that could be added to the discussion:

1)    The introduction does not provide sufficient background and more references are needed.

In the introduction, the authors mention TMJ arthritis as a serious disease. First, the distinction between arthritis and arthrosis should be included. Also, please explain why you believe that arthritis is serious? I highly recommend not using the term serious.

Are only patients with ADD are more suitable to CR? What about other types of DD?

Further information on CR, risk factors and characteristics of the patients are needed.

What is the rule of trauma if any in the development of CR?

You state (in order to formulate accurate and personalized treatment plan, analyse prognosis, and avoid potential bone resorption risk.) please explain how is that done?

Throughout the manuscript you mention the MRI is widely used to diagnose ADD, Pleas add information on the rule of clinical examination in the diagnosis of DD.

This sentence “The quality of condyle refers to its nutrient supply, metabolic activity and the potential of remodeling and regeneration.” is so vague pleas clarify.

2)    More elaborated description of Figure one is needed.

3)    Is the SIR measured for the condylar bone marrow and the cerebral cortex parenchyma? It is confusing because you keep referring to it differently through out the manuscript, consistency is required.

4)    The clinical implications of these findings need more discussion. Clinicians must know all the potential ones.

5)    What are the pitfalls and future recommendation of this study?

6)    The section “future prospect” is not clear and needs to be rewritten.

Author Response

Response to Reviewer 2 Comments

Dear reviewer:

  Many thanks for the insightful comments and suggestions of the referees. Those comments are all valuable and very helpful for revising and improving our paper, as well as the important guiding significance to our researches. About these comments, we have sepearated our responses point-by-point as follows.

Point 1:

  • The introduction does not provide sufficient background and more references are needed.
  • In the introduction, the authors mention TMJ arthritis as a serious disease. First, the distinction between arthritis and arthrosis should be included. Also, please explain why you believe that arthritis is serious? I highly recommend not using the term serious.
  • Are only patients with ADD are more suitable to CR? What about other types of DD?
  • Further information on CR, risk factors and characteristics of the patients are needed.
  • What is the rule of trauma if any in the development of CR?
  • You state (in order to formulate accurate and personalized treatment plan, analyse prognosis, and avoid potential bone resorption risk.) please explain how is that done?
  • Throughout the manuscript you mention the MRI is widely used to diagnose ADD, Pleas add information on the rule of clinical examination in the diagnosis of DD.
  • This sentence “The quality of condyle refers to its nutrient supply, metabolic activity and the potential of remodeling and regeneration.” is so vague pleas clarify.

Response 1:

  • We have already enriched the content and references in the introduction section such as further information on CR, risk factors and characteristics of the patients and the reason why patients with ADD tended to be at risk of CR while other types of DD didn’t show this tendency.
  • After discussion, we also considered that the sentence " TMJ arthritis is a serious disease " was not appropriate. We have already delete this sentence and rearranged the sequence of sentences hoping to make the article more logical.
  • According to our clinical observation, most patients with CR are always accompanied by ADD. The occurrence of other type of DD such as simple TMJ lateral disc displacement is rare which are usually asymptomatic and doesn’t require any treatment. We have also added this explanation to the introduction part.
  • Further information on CR, risk factors and characteristics of the patients have already been added into the introduction part.
  • Sorry, we didn’t get the meaning of “What is the rule of trauma if any in the development of CR?”. We are hoping to get further information.
  • In our study it was found that the condyle-to-cerebral cortex SIR was 1.07 ± 0.13 in patients with normal TMJ. For patients who have TMJ ADD but without CR, SIR decreased slightly to 1.03 ± 0.14. By contrast, SIR decreased widely to 0.78 ± 0.14 in patients with ADD and CR. Meanwhile according to ROC results, SIR ≥ 0.96 indicates healthy condyle, while SIR<0.96 is considered the optimal critical value for the diagnosis of CR. These findings can help clinicians estimate the status of patients' condyles by SIR Values. In our future expectation, we hope to rasie a “SIR degree” to formulate personalized treatment. When SIR>0.96, it is suggested that condyle is under a low risk of CR, and orthodontic treatment can be routinely performed. When 0.78<SIR<0.96, it should be careful that condyle is under a high risk of CR. At this stage, prevention of potential CR risk and health education are recommended. If the patient is undergoing orthodontic treatment, attention should be paid to the size and direction of orthodontic force to avoid exerting more pressure on the condyle. Meanwhile, the follow-up of the condyle by MRI is also important for early detection of CR. When SIR<0.78, it is indicated that CR has already happened or will certainly happen soon. At this stage, measures should be taken to delay the progression of CR, such as appropriate orthodontic treatment to relieve pressure on the condyle. Besides that, arthroplasty may be performed if necessary. It has been observed that arthroplasty can effectively slow the progression of CR. These ideas are what we think can provide advice and help for clinical work, and also the clinical implications of our study. The response above has also been added to the future prospects part of the article.
  • In clinical examination, disc displacement is often diagnosed by examining the patient's mouth opening degree and whether there exists clicking or limitation when opening mouth. We have added this information in the introduction part.
  • The sentence “The quality of condyle refers to its nutrient supply, metabolic activity and the potential of remodeling and regeneration.” seems a little bit vague indeed. However it requires a lot of basic experimental theory to explain this point, which is inconsistent with the focus of this paper, so we decided to delete it at last.

Point 2: More elaborated description of Figure one is needed.

Response 2: We have added more description of Figure one and made the corresponding identification on the figure.

Point 3: Is the SIR measured for the condylar bone marrow and the cerebral cortex parenchyma? It is confusing because you keep referring to it differently through out the manuscript, consistency is required.

Response 3: In our study, SIR means the signal intensity ratios between condyle and cerebral cortex. We have already made it consistent in the paper.

Point 4: The clinical implications of these findings need more discussion. Clinicians must know all the potential ones.

Response 4:We have already added clinical implications of SIR which have been mentioned above in point 1 into the first paragragh of future prospect part.

Point5:  What are the pitfalls and future recommendation of this study?

Response 5:The pitfalls of this study may be the lack of enough cases. And these findings are lack of the support of pathological basis of condyle. These pitfalls will be our future work priorities. The future recommendation of this study is to raise a SIR degree in ADD patients which will help clinicians make the correct decision and avoid the risk of CR at different stage. Besides that, we will introduce AI technology in MRI to assist the clinicians to judge the status of condyle.

Point6: The section “future prospect” is not clear and needs to be rewritten.

Response 6: We have already enriched the “future prospect” section and made it much easier to undesrstand.